# Synthesis of an aromatic N-heterocycle derived from biomass and its use as a polymer feedstock

Yu Qi[1], Jinyan Wang[1], Yan Kou[2], Hongchang Pang[3], Shouhai Zhang[1], Nan Li[1], Cheng Liu [1], Zhihuan Weng[1] & Xigao Jian[1]

Aromatic N-heterocyclic compounds are very important chemicals, which are currently produced mostly from petroleum. Here we report that a pyridazine-based compound 6-(4-hydroxy-3-methoxyphenyl)pyridazin-3(2H)-one (GSPZ) can be efficiently synthesized by the Friedel-Crafts reaction of guaiacol and succinic anhydride, both of which can be derived from biomass. GSPZ is then treated with bio-based epichlorohydrin to prepare the epoxy resin precursor GSPZ-EP. With 4,4'-diaminodiphenylmethane as curing agent, GSPZ-EP possesses higher glass transition temperature (187 °C vs. 173 °C) and shows a 140%, 70 and 93% increase in char yield (in $N_2$), storage modulus (30 °C) and Young's modulus, respectively when compared with a standard petroleum-based bisphenol A epoxy resin. Moreover, the cured GSPZ-EP shows good intrinsic flame retardancy properties and is very close to the V-0 rating of UL-94 test. This work opens the door for production of aromatic N-heterocyclic compounds, which can be derived from biomass and employed to construct high performance polymers.

[1] State Key Laboratory of Fine Chemicals, Liaoning High Performance Resin Engineering Research Center, Department of Polymer Science & Engineering, Dalian University of Technology, Dalian 116024, China. [2] Thermochemistry Laboratory, Liaoning Province Key Laboratory of Thermochemistry for Energy and Materials, Dalian National Laboratory for Clean Energy, Dalian Institute of Chemical Physics, Chinese Academy of Sciences, Dalian 116023, China. [3] School of Chemical Engineering, Dalian University of Technology, Dalian 116024, China. Correspondence and requests for materials should be addressed to Z.W. (email: zweng@dlut.edu.cn)

Biomass is a widely available alternative feedstock for producing useful chemical compounds and can alleviate the reliance on finite petroleum-based resources[1]. The worldwide energy crisis and related environmental problems have provoked extensive research and development programs, involving biomass conversion[2–7]. Using the knowledge derived from the petroleum industry, several key known compounds or polymers have been made using biomass, such as ketones[8], monoterpenes[9], diethyl terephthalate[10], meta-xylylenediamine[11], phthalic anhydride[12], poly (ethylene-2,5-furandicarboxylate) (PEF)[13], polycarbonate[14], poly-lactic acid (PLA)[15], and polyethylene terephthalate (PET)[16].

Aromatic N-heterocyclic compounds are very important chemicals, as their motifs are found in many pharmaceuticals, natural products, and functional materials. Unfortunately, their synthesis from biomass-derived starting materials remains challenging and is still very limited, owing to the lack of efficient methods for biomass-derived precursors[17,18]. Bhusal et al. have shown that a suite of diverse N-heterocycles can be prepared from dimethyl itaconate and pyrrole, two compounds attainable from biomass[19]. Therein, the pyrrole was produced from furan and ammonia, and the reaction condition for preparing N-heterocylces was relatively harsh and difficult to scale up. Kallmeier et al. reported on the base-metal-catalyzed synthesis of pyrroles from secondary alcohol and amino alcohol starting materials via a combination of catalytic dehydrogenation and condensation steps[20]. However, in that case, the prepared pyrroles were only partially bio-based, as just the secondary alcohols can be obtained from indigestible and abundantly available lignocellulose biomass. Therefore, it is particularly important to further develop facile synthetic routes to bio-based aromatic N-heterocycles[21].

To date, one of the challenges for bio-based polymers is to show complementary or improved properties compared with polymers that are currently available[22,23]. Furthermore, it has been shown that introduction of aromatic heterocycles into polymers can improve their properties or impart new functionality[24–26]. Given these facts, developing high-performance N-heterocycle polymers from biomass is an important area for further research. Despite this, reports on preparation of bio-based monomers derived from aromatic N-heterocycles suitable for polymerization are particularly rare[27,28].

In this work, we offer a universal protocol that relies on Friedel–Crafts acylation of the phenolic compound with anhydride, followed by cyclization and dehydrogenation with hydrazine hydrate and sodium 3-nitrobenzenesulfonate, respectively, to afford a pyridazine-based aromatic N-heterocycle, comprising only biomass carbon. During the procedure, all three synthetic steps are facile and straightforward. Moreover, the products are easily isolated without utilizing column chromatography, implying excellent potential for scale-up or commercial applications. The prepared bisphenol-like compound can be employed as a monomer with a rigid aromatic structure to enhance the properties of polymers. The methoxy moiety on the ortho site to phenolic hydroxyl in the as-prepared aromatic N-heterocycle can significantly decrease the binding affinity to endocrine compounds, whereas bisphenol A is considered as an endocrine disruptor because it can bind to a variety of receptors in biological systems[29–32].

In this study, we also describe access to a fully renewable diepoxy monomer from the above-mentioned aromatic N-heterocycle. Following its treatment with epichlorohydrine and then using 4,4'-diaminodiphenylmethane (DDM) as a curing agent, the resulting bio-based epoxy resin shows superior properties to its petroleum-based counterpart (a standard bisphenol A epoxy resin DGEBA). These include a higher glass transition temperature, better thermal mechanical properties, and excellent intrinsic flame retardancy, thanks to the introduction of the pyridazine-based structure. This work affords both a facile synthetic toolbox for sustainable production of aromatic N-heterocycles and a method to improve the competitiveness of bio-based polymers relative to their petroleum-based counterparts.

## Results

**Synthetic strategy**. The fully bio-based aromatic N-heterocycle 6-(4-hydroxy-3-methoxyphenyl)pyridazin-3(2H)-one (GSPZ) was synthesized from guaiacol and succinic anhydride in three steps depicted in Fig. 1: (i) guaiacol was coupled by succinic anhydride through Friedel–Crafts acylation to yield an intermediate acid 3-(4-hydroxy-3-methoxybenzoyl)propionic acid (GSA); (ii) the cyclization reaction of GSA with hydrazine hydrate generated an N-heterocyclic compound 6-(4-hydroxy-3-methoxyphenyl)-4,5-dihydro-2H-pyridazin-3-one (GSHZ); (iii) the GSHZ was subsequently dehydrogenized to obtain the target compound GSPZ. Furthermore, the bisphenol-like monomer GSPZ can be employed to synthesize a bio-based epoxy precursor GSPZ-EP by reaction of GSPZ with excess epichlorohydrin (produced via bio-based glycerol chlorination[33]), in the presence of a catalyst. All the reaction steps were facile and straightforward. Moreover, the

**Fig. 1** A sustainable route to produce bio-based pyridazine-based compounds. Reaction conditions: **a** guaiacol, succinic anhydride, aluminum chloride anhydrous, $C_2H_4Cl_2$, 30 ºC, 8 h. **b** $N_2H_4 \cdot H_2O$, $H_2O$, reflux 2.5 h. **c** Sodium 3-nitrobenzenesulfonate, 0.5 mol·L$^{-1}$ NaOH aqueous solution, reflux 10 h. **d** Epichlorohydrin, benzyltriethylammonium chloride, 80 ºC, 3 h, followed by adding NaOH, 80 ºC, 1 h

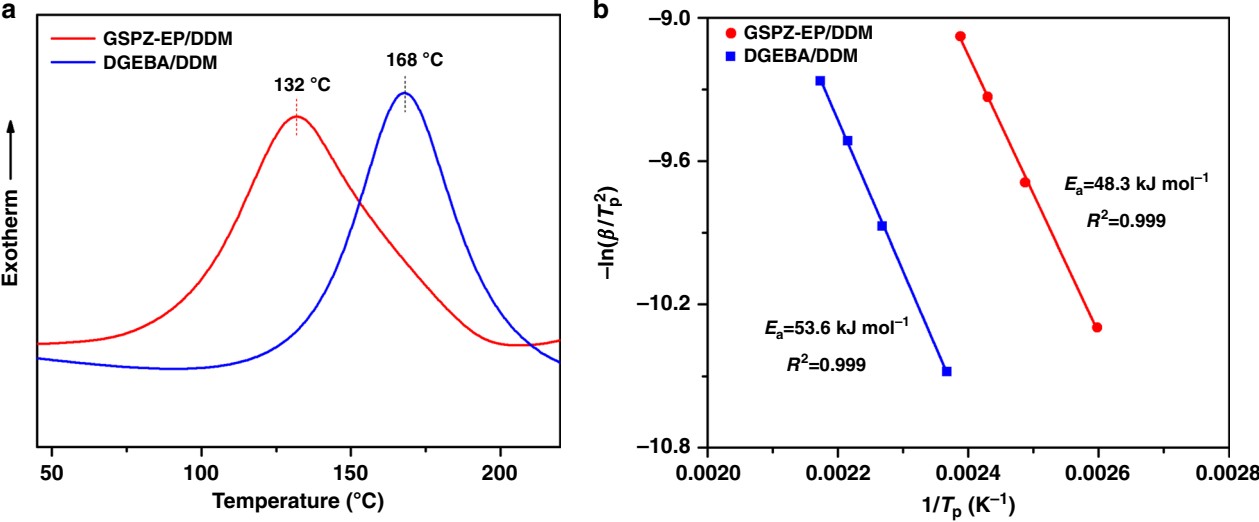

**Fig. 2** Evaluation on the curing activity of GSPZ-EP/DDM and DGEBA/DDM. **a** Non-isothermal thermographs of GSPZ-EP/DDM and DGEBA/DDM with the heating rate of 10 K·min$^{-1}$. **b** Linear plots of $-\ln(\beta/T_p^2)$ versus $1/T_p$ based on Kissinger's equation of GSPZ-EP/DDM and DGEBA/DDM

products were easily isolated without column chromatography. This implies significant potential for scaling up industrial applications. The structures of the prepared bio-based compounds were characterized using a variety of techniques, as shown in Supplementary Figs. 1–5. In addition, the ESI-MS spectrum of GSPZ-EP shows that [M + H$^+$] = 331.13, [M + Na$^+$] = 353.11, which is consistent with the theoretical value. The epoxy value of GSPZ-EP is 0.58 mol·100 g$^{-1}$ determined by ASTM D1652. All these results fully confirm that GSPZ-EP with a designed structure has been facilely synthesized.

**Curing behavior of the GSPZ-EP/DDM system**. The non-isothermal curing behaviors of the bio- and petroleum-based epoxy resin systems were shown in Fig. 2a with DDM as a curing agent, because its melting point (90 °C) is similar to GSPZ-EP (92 °C), which is more conductive to the operation of the curing reaction. It can be seen that both the systems showed only a single exothermic peak without any melting peaks or additional exothermic peaks. This may indicate that GSPZ-EP and DDM could form a eutectoid in molecular homogeneity after melt blending. Compared with DGEBA/DDM, GSPZ-EP/DDM showed lower curing exothermic peak. According to the literature[34,35], the temperature value of the curing exothermic peak is often used to estimate the curing reactivity. A lower exothermic temperature value indicates a more reactive system. Thus, the lower curing exothermic peak of GSPZ-EP/DDM indicated that it had a higher curing reactivity than DGEBA/DDM. Furthermore, the curing apparent activation energy calculated based on Kissinger's method[36] (Supplementary Note 1) was illustrated in Fig. 2b (the curing kinetic curves can be seen in Supplementary Fig. 6), and GSPZ-EP/DDM presented lower $E_a$ (48.3 kJ·mol$^{-1}$) than DGEBA/DDM (53.6 kJ·mol$^{-1}$), providing further evidence that GSPZ-EP/DDM had a higher curing reactivity than DGEBA/DDM. The higher curing reactivity of GSPZ-EP/DDM may be due to the presence of the tertiary amine functional group. These are an effective class of a catalytic curing agent widely used to initiate polymerization of various epoxy resins[37–39]. The tertiary amine structure in GSPZ-EP could also promote the curing of the GSPZ-EP/DDM system; therefore, GSPZ-EP/DDM showed a higher curing reactivity than DGEBA/DDM under the same conditions. Viscosity is an important parameter for epoxy resin; upon heating to 110 °C, the bulk viscosity for GSPZ-EP was 203 mPa·s, whereas it was 41 mPa·s in the case of DGEBA.

**Self-curing behavior of GSPZ-EP**. As mentioned above, a tertiary amine represents a class of effective catalytic curing agents for epoxy resins; therefore, we employed TGA-DSC simultaneous thermal analysis to track the reaction process without a curing agent (Fig. 3a). GSPZ-EP showed a clear melting endotherm at 92 °C, significantly higher than room temperature, making the system stable at room temperature and convenient for storage and transport. Surprisingly, it also showed two obvious exotherms in the DSC curve, while there were no obvious thermal events on DGEBA treated by the same process. For the TGA curve, the mass of GSPZ-EP didn't show significant decline during the first two exotherms, while the third exotherm was accompanied with mass reduction. This indicated that the first two exotherms were due to curing, while the third exotherm was caused by decomposition. For further verification, we used rheological measurement to monitor viscosity changes of GSPZ-EP at elevated temperatures. As illustrated in Fig. 3b, the lowest viscosity of melted GSPZ-EP was 0.03 Pa·s, a value beneficial for processing. The viscosity remained approximately constant until 230 °C where an abrupt increase was observed, that indicated that the self-curing phenomenon had occurred. To the best of our knowledge, the ability of epoxy resins derived from biomass to show self-curing characteristics has seldom been previously reported.

To better understand the self-curing reaction of GSPZ-EP, non-isothermal curing kinetics was carried out on DSC. Figure 3c showed the DSC curves of GSPZ-EP with four different heating rates. All four curves exhibited two curing exotherms, suggesting that the curing reaction may have two reaction types. The third exothermic peak shown in curves is probably caused by decomposition and therefore does not merit any further discussion here. The curing apparent activation energy was calculated from the slope of kinetic plots according to the Kissinger's method (Supplementary Fig. 7). The apparent activation energy of the first and the second exothermic peaks were 84.8 kJ·mol$^{-1}$ and 86.6 kJ·mol$^{-1}$, respectively, falling within the range of 81–93 kJ·mol$^{-1}$ reported by Barton et al.[40] who used imidazoles as curing agents for epoxy resin. Thus, the curing mechanism of GSPZ-EP may be similar to that of imidazoles, as they not only show similar curing activation energy[41], but also contain similar tertiary amines. In addition, most epoxy resins cured with imidazoles also display two curing exothermic peaks[42–44]. In agreement with the previous literature[45,46], we

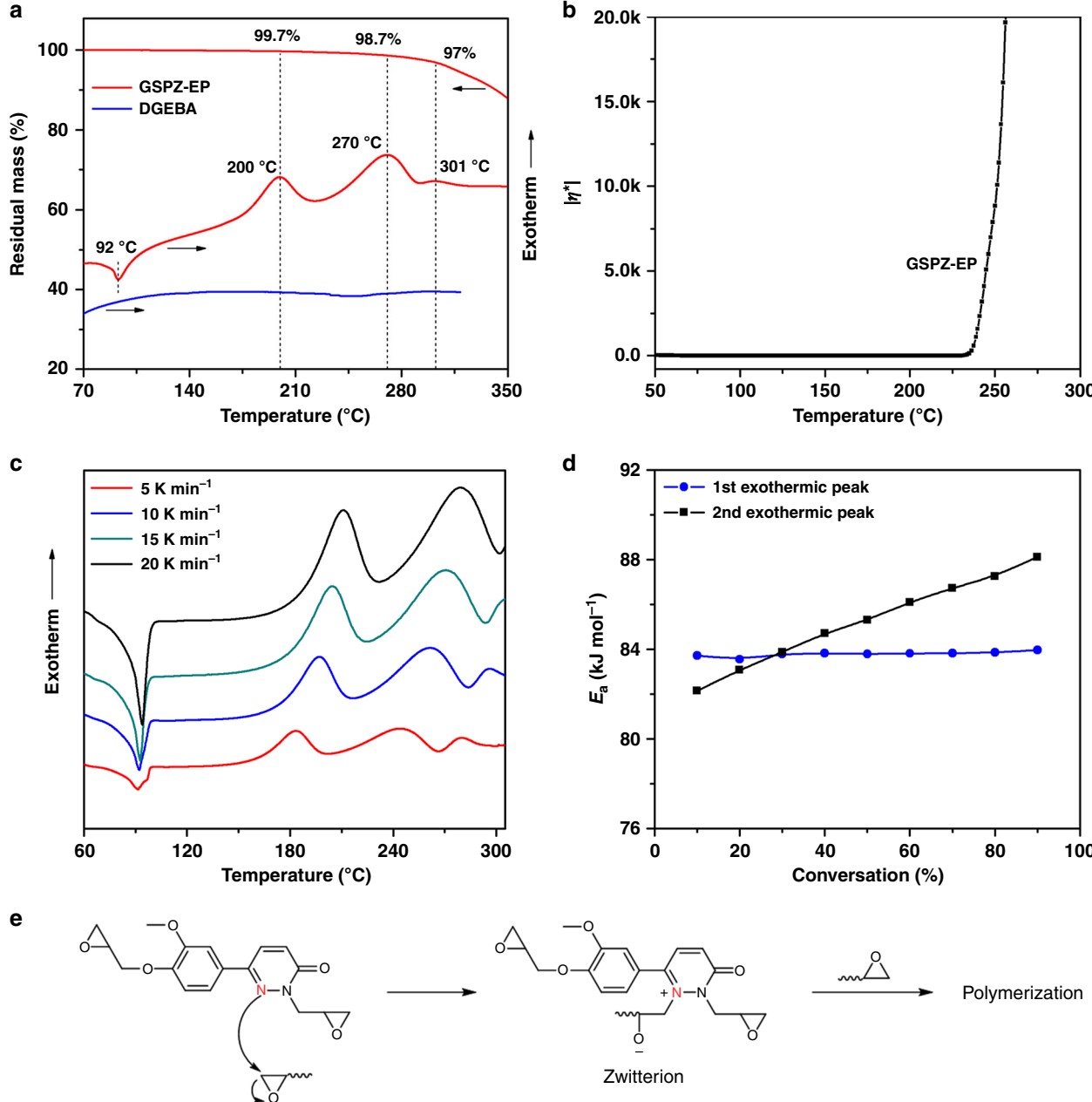

**Fig. 3** Analysis of the self-curing behavior and the proposed self-curing mechanism of GSPZ-EP. **a** TGA-DSC simultaneous thermal analysis curves of GSPZ-EP and non-isothermal DSC curves of DGEBA; **b** complex viscosity ($\eta^*$) of GSPZ-EP as a function of temperature at a heating rate of 3 °C·min⁻¹; **c** non-isothermal curing kinetic curves of GSPZ-EP with a heating rate of 5, 10, 15, and 20 K·min⁻¹; **d** variation of activation energy vs. conversion for the first and the second exothermic peaks; **e** possible reaction mechanism of self-curing GSPZ-EP

propose the mechanism for self-curing GSPZ-EP as depicted in Fig. 3e. The reaction involves two steps: initially the 3-position nitrogen attacks and opens an epoxide ring to form a zwitterion and then the homopolymerization of the GSPZ-EP occurs.

Isoconversional analysis is an effective method to reveal the curing reaction mechanism in detail[47]. The activation energy can be calculated using integral isoconversion, such as the Friedman, Ozawa, Flyn, and Wall methods[48]. According to the ICTAC kinetic project[49], it is recommended to use Starink methods (shown in Supplementary Note 2) due to their higher accuracy. Figure 3d exhibits $E_a$–$\alpha$ relation curves for the first and second exothermic peaks (the calculative process is shown in Supplementary Figs. 8, 9). For the first exothermic peak, the activation energy remained constant for the entire range of conversion;

similar results were also observed by Leena et al.[42]. For the second exothermic peak, the activation energy increased along with the percentage conversion. This is also in agreement with previous reports[50,51], describing an epoxy resin exposed to self-curing. It is consistent with a homopolymerization forming a three-dimensional network. During the course of the reaction, the steric hindrance and viscosity increase, as the network is formed requiring a higher activation energy as the reaction continues. This interpretation of the isoconversional analysis data further verifies that the mechanism reported in Fig. 3e is reasonable.

**Thermal stability of the GSPZ-EP/DDM system.** The thermal stability and degradation behaviors of the cured epoxies were investigated by TGA in N₂ and air atmosphere. As illustrated in

| Table 1 TGA and DTG data of the cured resins | | | | | | | | |
|---|---|---|---|---|---|---|---|---|
| **Samples** | $T_{d5\%}$ (°C) | | $C_{yd700}$ (%) | | $T_{max}$ (°C) | | $R_{max}$ (%·min$^{-1}$) | |
| | **N₂** | **Air** | **N₂** | **Air** | **N₂** | **Air** | **N₂** | **Air** |
| GSPZ-EP/DDM | 331 | 316 | 42.3 | 1.0 | 368.3 | 359, 586.7 | 0.15 | 0.008, 0.1 |
| DGEBA/DDM | 378.7 | 367.3 | 17.8 | 0.5 | 399 | 393, 589.3 | 0.34 | 0.3, 0.05 |

Supplementary Fig. 10 and Table 1, compared with DGEBA/DDM, GSPZ-EP/DDM showed lower 5 wt% mass loss temperature under both atmospheres. That may be attributed to the dissociation of the methoxyl groups from guaiacol moieties[52]. Nonetheless, the initial decomposition temperature of GSPZ-EP/DDM was still significantly higher than its glass transition temperature ($T_g$), showing satisfactory thermal stability for practical applications. In a N₂ atmosphere, both GSPZ-EP/DDM and DGEBA/DDM showed a one-step degradation process. The maximum degradation rate of GSPZ-EP/DDM was far lower than DGEBA/DDM, by a factor of about 55.9%. More surprisingly, GSPZ-EP/DDM showed high char yield of 42.3% at 700 °C, which is 1.4 times higher than that of DGEBA/DDM. Such a high char yield is unusual in epoxy resin systems without any flame retardant.

Conversely, the main degradation processes of GSPZ-EP/DDM and DGEBA/DDM in air occurred in two stages. The first degradation stage is likely due to the thermal oxidative degradation of the cross-linked network. In air, GSPZ-EP/DDM exhibited apparently lower thermal degradation rates than DGEBA/DDM, by 97.3%. After the first stage, GSPZ-EP/DDM presented more charred residue than DGEBA/DDM. The second degradation process was probably due to further oxidative degradation of the charred residue[53]. GSPZ-EP/DDM showed a much higher 50 wt% loss temperature than DGEBA/DDM (545 °C vs. 427 °C), which may indicate that the char residue of GSPZ-EP/DDM was much stronger. When the temperature was further increased, the residue char burned out, leaving almost no char residue above 700 °C.

The excellent thermal behavior of GSPZ-EP/DDM with an extremely high char yield and slow degradation rate, may be explained by its rigid fully aromatic structure containing tertiary amine. As reported in the literature[54], compounds containing tertiary amine groups have favorable char formation and thermal stability properties. Also, the highly compact fully aromatic structure could also promote charring[55]. The higher char residue of GSPZ-EP/DDM could serve as a thermal insulating layer, protecting the matrix from further degradation. In addition, the high char yield of GSPZ-EP/DDM could also promote the flame retardancy of the resin, as discussed below.

**Flame retardancy of cured epoxy resins**. First, microscale combustion calorimetry (MCC) was used to evaluate the flammability of the cured epoxy resins. The heat release rate (HRR) versus temperature curves and the related data are shown in Fig. 4. It is clear that both systems show no obvious heat release below 300 °C, which indicates that no combustible substances were generated in that temperature range. Above 300 °C, GSPZ-EP/DDM released heat at a lower temperature than DGEBA/DDM, indicating that GSPZ-EP/DDM starts to decompose at lower temperature (363.4 °C vs. 390.1 °C), which is in accordance with the results from TGA. Significantly, GSPZ-EP/DDM shows a much lower peak heat release rate (PHRR) and total heat release (THR) than DGEBA/DDM and the reduction is about 60.5% (184.5 W·g$^{-1}$ vs. 467.5 W·g$^{-1}$) and 54.4% (12.4 kJ·g$^{-1}$ vs. 27.2 kJ·g$^{-1}$), respectively. These results indicate that GSPZ-EP/DDM

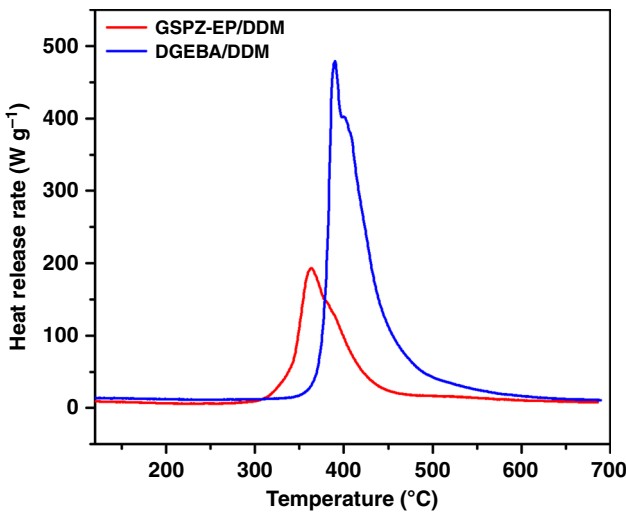

**Fig. 4** The flammability of the cured resins evaluated by the MCC test. Heat release rate (HRR) against temperature curves from MCC tests for cured GSPZ-EP/DDM and DGEBA/DDM resins

produces less combustible substances than DGEBA/DDM during the test process, suggesting that GSPZ-EP/DDM shows far less flammability than DGEBA/DDM.

Second, the vertical burning test (UL-94) was employed as an important method to estimate the flame retardancy of polymeric materials. Representative digital photos of the resins taken during the burning process are shown in Fig. 5a–c. After the first ignition, DGEBA/DDM burned fiercely, showing no self-extinguishing phenomenon. Even after 96 s, it was still burning with melt-dripping, clearly indicating that DGEBA/DDM is highly flammable, more than any classification on the UL-94 test. In contrast, when GSPZ-EP/DDM was first ignited for 10 s, it could self-extinguish after 5.3 s. Following a second ignition, after 7 s, it still could self-extinguish without any melt-dripping. These results indicated that the intrinsic flame retardancy property of cured GSPZ-EP/DDM was good and very close to the UL-94 rating of V-0. After the UL-94 test, the residual test bar of DGEBA/DDM (shown in Fig. 5d) was very short and contained almost no char residue, due to its poor charring ability, which was consistent with the TGA result. Conversely, the tested bar of GSPZ-EP/DDM post combustion (shown in Fig. 5e) was still long and covered with intumescent char layers. These combined results indicate that GSPZ-EP/DDM shows a far superior flame retardancy to DGEBA/DDM, which may be attributed to the high charring ability aroused by the full aromatic structure with tertiary amine in former's network[56,57], and the proposed mechanism will be illustrated in our further work.

**Thermomechanical and mechanical properties of the resins**. Generally, epoxy resins with excellent flame retardancy are inherently poor in other properties, such as glass transition temperature and mechanical properties[58,59]. Conversely, GSPZ-EP/DDM shows excellent flame retardancy because of its intrinsic structure characteristic without sacrificing other properties.

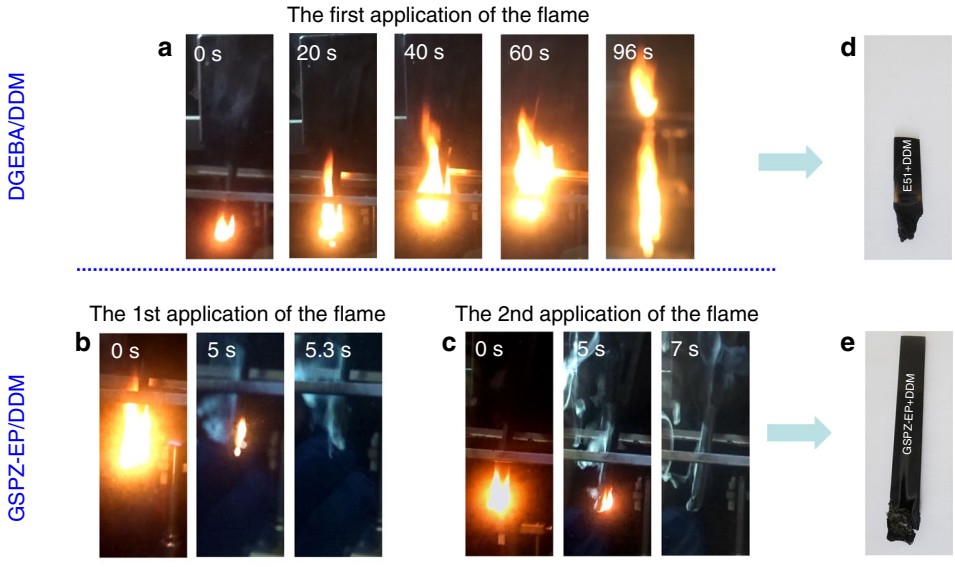

**Fig. 5** The flame retardancy of the cured resins evaluated by the UL-94 test. **a–c** are video screenshots of the cured DGEBA/DDM and GSPZ-EP/DDM resins during the vertical burning test: **a** The cured DGEBA/DDM resin was applied to the first ignition at the bottom for 10 s. After the ignition, DGEBA/DDM couldn't self-extinguish until burned to the clamp. The burning process was recorded by the video, and the combustion phenomena of 0, 20, 40, 60, and 96 s after the first 10-s ignition, were captured from the video and are exhibited. **b** The cured GSPZ-EP/DDM resin was applied to the first ignition at the bottom for 10 s. After the ignition, GSPZ-EP/DDM could self-extinguish in a short time. The combustion phenomena of 0, 5, and 5.3 s after the first 10-s ignition were captured from the video. **c** The cured GSPZ-EP/DDM resin was applied to the second 10-s ignition at the bottom immediately, when it self-extinguished after the first ignition. The combustion phenomena of 0, 5, and 7 s after the second ignition were captured from the video and were exhibited. **d** The digital photo of DGEBA/DDM test bar after the vertical burning test. **e** The digital photo of GSPZ-EP/DDM test bar after the vertical burning test

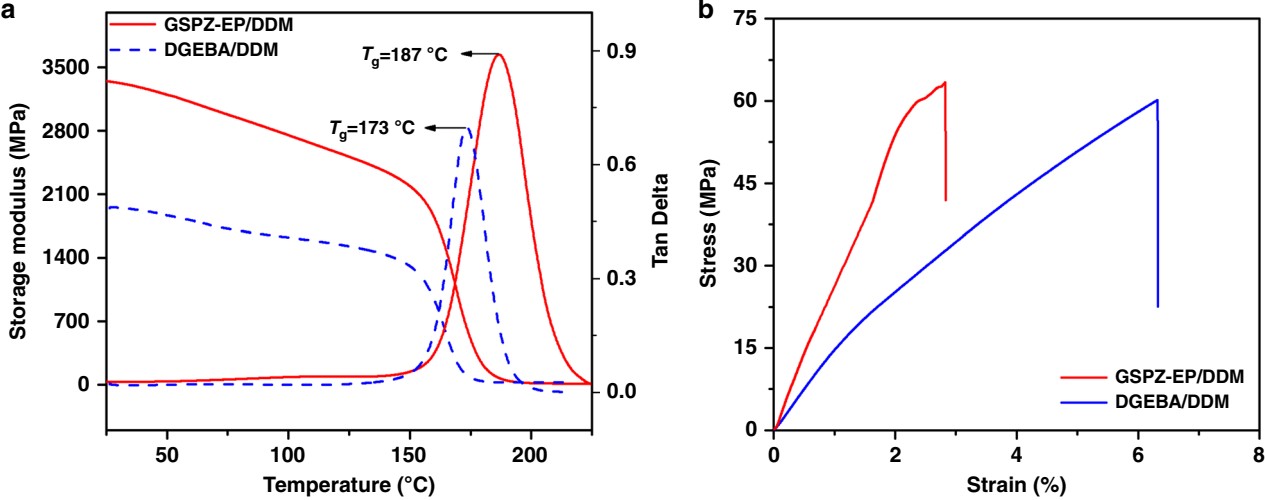

**Fig. 6** The thermomechanical and mechanical properties of the cured resins. (**a**) Dynamic mechanical properties spectra for storage modulus and tan δ against temperature curves of GSPZ-EP/DDM and DGEBA/DDM. (**b**) Tensile stress–strain curves of GSPZ-EP/DDM and DGEBA/DDM

We first employed dynamic mechanical analysis (DMA) to study the thermomechanical properties of cured GSPZ-EP/DDM and DGEBA/DDM in a comparable way. Figure 6a shows the corresponding storage modulus and tan δ against temperature curves. Accordingly, the obtained storage modulus, glass transition temperature, and cross-linking density are summarized in Table 2. From Fig. 6a, we can see that the storage modulus of GSPZ-EP/DDM is evidently higher than DGEBA/DDM over the entire temperature range. Particularly at 30 °C, GSPZ-EP/DDM shows extremely high storage modulus of 3327 MPa, which is 70.4% higher than DGEBA/DDM. Storage modulus is a parameter that reflects the rigidity of resins directly. The fully

aromatic structure of GSPZ-EP helps endow the system with extraordinary rigidity. In addition, the existence of nitrogen may facilitate the formation of hydrogen bonds and strengthen the intermolecular forces, which further enhance the storage modulus of the system.

The glass transition temperature is an important parameter for thermosetting resins. It determined the maximum working temperature of resin systems. Here, the glass transition temperature was obtained by the peak value temperature of the tan δ vs. temperature curves. As shown in Fig. 6a, GSPZ-EP/DDM possesses a relatively high $T_g$ of 187 °C, compared with DGEBA/DDM with a $T_g$ of 173 °C. Generally, $T_g$ depends on the

**Table 2 Key parameters summarized from DMA and tensile testing for cured resins**

| Sample | $E'$ (30 ºC) [a] (MPa) | $T_g$[b] (ºC) | $V_r$[c] (mol·m$^{-3}$) | Young's modulus (MPa) | Ultimate tensile strength (MPa) | Elongation at break (%) |
|---|---|---|---|---|---|---|
| GSPZ-EP/DDM | 3327 ± 38 | 187 | 1200 ± 68 | 2798 ± 120 | 63.5 ± 5 | 2.8 ± 0.5 |
| DGEBA/DDM | 1952 ± 20 | 173 | 2432 ± 46 | 1452 ± 108 | 61.7 ± 3 | 6.3 ± 0.3 |

[a]$E'$(30 ºC): storage modulus at 30 ºC measured by DMA
[b]$T_g$: glass transition temperature defined by the peak value temperature of tanδ against temperature curve
[c]$V_r$: cross-linking density

combined effects on both the rigidity of the chain structure and the cross-linking density of the resins. The aromatic structure enhances the rigidity of the GSPZ-EP/DDM system, and restricts the molecular chain mobility, which is in favor of improving the $T_g$ of the system. While the cross-linking density of GSPZ-EP/DDM calculated according to the previous report[60] was lower than DGEBA/DDM (Table 2), ultimately, GSPZ-EP/DDM shows a higher $T_g$ than DGEBA/DDM, because the higher rigidity effect is dominant.

Figure 6b gives the tensile stress–strain curves of the cured resins, and the corresponding tensile properties are summarized in Table 2. As shown in Fig. 6b, both of the resins present a brittle fracture without yielding. GSPZ-EP/DDM shows a much higher Young's modulus of 2798 MPa, which is 93% higher than that of DGEBA/DDM. This is in agreement with the storage modulus values and further reflects the rigidity of the resin. The more rigid fully aromatic structure and stronger intermolecular forces originated by the existence of polar C–N bonds improving the Young's modulus of the GSPZ-EP/DDM system, at the same time restricting the mobility of the molecular chain, makes the GSPZ-EP/DDM system more brittle. Thus, it leads to its lower elongation at break than DGEBA/DDM. Although GSPZ-EP/DDM possesses a more rigid structure and stronger intermolecular forces, it doesn't show an obvious improvement in ultimate tensile strength. That is because the cross-linking density of GSPZ-EP/DDM is much lower than that of DGEBA/DDM. However, the ultimate tensile strength of GSPZ-EP/DDM is still very high.

## Discussion

In this work, we opened up a way to synthesize a fully bio-based aromatic N-heterocyclic compound, GSPZ. The compound can be used as a precursor to synthesize a fully bio-based epoxy resin, GSPZ-EP. The fully aromatic N-heterocyclic structure of GSPZ-EP imparts a series of excellent properties. With DDM as a curing agent, GSPZ-EP shows a higher curing reactivity than the petroleum-based counterpart DGEBA, thanks to a tertiary amine group. In addition, the cured GSPZ-EP/DDM shows extremely high char yield of 42.3% at 700 ºC in $N_2$, while the value of DGEBA/DDM was only 17.8% under identical conditions. Interestingly, the cured GSPZ-EP/DDM showed excellent intrinsic flame retardancy. Compared with DGEBA/DDM, GSPZ-EP/DDM showed a 60.5 and 54.4% decrease in exothermal release rate and total heat release via the MCC test. Furthermore, it was very close to the UL-94 rating of V-0, while DGEBA/DDM passed no rating. Unlike many flame-retardant materials, GSPZ/DDM shows excellent structural properties, such as a high glass transition temperature (187 ºC), a storage modulus of 3327 MPa (30 ºC), and a tensile modulus of 2798 MPa. These values are substantially higher than those of DGEBA/DDM. After further development, GSPZ-EP shows an excellent potential to replace DGEBA in practical applications.

## Methods

**Materials**. Guaiacol was purchased from Sinopharm Chemical Reagent Co., Ltd, China. Succinic anhydride and benzyltriethylammonium chloride were purchased from Saen Chemical Technology Co., Ltd., Shanghai, China. 4,4′-methylenedianiline was obtained from Mackliin Biochemical Co., Ltd., Shanghai, China. Epichlorohydrin was supplied by Aladdin-reagent Co., Ltd., Shanghai, China. Aluminum chloride anhydrous, hydrazine hydrate, 1,2-dichloroethane, sodium hydroxide, and sodium sulfate anhydrous were purchased from Damao Chemical Reagent Factory, Tianjin, China. Epoxy resin (DGEBA, trade name E51, epoxy value 0.51 mol·100 g$^{-1}$) was purchased from Nantong Xingchen Synthetic Materials Co., Ltd, China. All chemicals and solvents were used without further purification.

**Synthesis of the intermediate acid GSA**. Guaiacol (31.0 g, 0.25 mol) was dissolved in 380 mL of dry dichloroethane at 0 ºC. Then, anhydrous aluminum chloride (83.3 g, 0.625 mol) was slowly added into this system. The reaction mixture was maintained at 0 ºC with continuous stirring for 20 min, and then succinic anhydride (25.0 g, 0.25 mol) was slowly added. After continuous stirring for another 30 min at 0 ºC, the system was warming up to 30 ºC and stirred at that temperature for 8 h. Afterward, the reaction was quenched with ice-cold water, followed by acidification with 20 mL of concentrated HCl. The precipitate was collected on a filter, and recrystallized from water to give 3-(4-hydroxy-3-methoxybenzoyl)propionic acid (GSA) as white crystals (25.4 g, yield 45.4%). mp: 158 ºC. $^1$H-NMR (400 MHz, DMSO-d$_6$, ppm): δ = 2.52 (t, 2 H), 3.15 (t, 2 H), 3.81 (s, 2 H), 6.85 (d, 1 H), 7.44 (d, 1 H), 7.52 (dd, 1 H), 9.95 (s, 1 H), and 12.07 (s, 1 H). $^{13}$C-NMR (100 MHz, DMSO-d$_6$, ppm): δ = 27.96, 32.47, 55.56, 111.00, 114.92, 122.74, 128.33, 147.46, 151.62, 173.88, and 196.47. FT-IR (KBr, cm$^{-1}$): 3450 (O–H), 1712, and 1674 (C = O).

**Synthesis of the aromatic N-heterocycle GSHZ**. GSA (22.4 g, 0.1 mol) was reacted with 80% hydrazine hydrate (15.6 g, 0.25 mol) in 150 mL of distilled water at reflux for 2.5 h. Afterward, the reaction mixture was cooled down and the precipitates were filtered, washed with water, and dried to get 6-(4-hydroxy-3-methoxyphenyl)-4,5-dihydro-2H-pyridazin-3-one (GSHZ) as light yellow crystals. (21 g, yield 95%), mp: 210.7 ºC. $^1$H-NMR (400 MHz, DMSO-d$_6$, ppm): δ = 2.40 (t, 2 H), 2.89 (t, 2 H), 3.79 (s, 3 H), 6.80 (d, 1 H), 7.16 (dd, 1 H), 7.34 (d, 1 H), 9.38(s, 1 H), and 10.75 (s, 1 H). $^{13}$C-NMR (100 MHz, DMSO-d$_6$, ppm): δ = 21.75, 26.09, 55.46, 108.90, 115.02, 119.33, 127.21, 147.56, 148.12, 149.49, and 166.98. FT-IR (KBr, cm$^{-1}$): 3215, 3095 (N–H, O–H), and 1645 (C = O).

**Synthesis of the aromatic N-heterocycle GSPZ**. GSHZ (11 g, 0.05 mol), sodium 3-nitrobenzenesulfonate (14.1 g, 0.0625 mol), and sodium hydroxide (11 g, 0.275 mol) were added in 550 mL of distilled water and the mixture was refluxed for 10 h. Then, the reaction was cooled down to room temperature, acidified with concentrated HCl. The precipitate was collected by filtration and washed several times with water, and the final product 6-(4-hydroxy-3-methoxyphenyl)pyridazin-3(2 H)-one was obtained with 51% of yield (5.6 g). mp: 244.7 ºC. $^1$H-NMR (400 MHz, DMSO-d$_6$, ppm): δ = 3.83 (s, 3 H), 6.85 (d, 1 H), 6.94 (d, 1 H), 7.29 (dd, 1 H), 7.40 (d, 1 H), 8.00(d, 1 H), 9.40 (s, 1 H), and 13.00 (s, 1 H). $^{13}$C-NMR (100 MHz, DMSO-d$_6$, ppm): δ = 56.04, 109.68, 116.03, 119.27, 126.32, 130.33, 131.80, 144.36, 148.41, and 160.58. FT-IR (KBr, cm$^{-1}$): 3435, 3116 (N–H, O–H), and 1655 (C = O).

**Synthesis of GSPZ-EP**. GSPZ (5.45 g, 25 mmol), epichlorohydrin (46.26 g, 500 mmol), and benzyltriethylammonium chloride (BTEAC, 0.57 g, 2.5 mmol) as the catalyst were placed in a three-necked round-bottomed flask with a mechanical stirrer, a reflux condenser, and an inlet for nitrogen. The mixture was vigorously stirred at 80 ºC for 3 h. Then, aqueous sodium hydroxide (40% w/w, 3.75 mL) was added dropwise to the system and stirred for 1 h at 80 ºC. Afterward, the mixture was cooled down to room temperature and transferred to a separating funnel. The resulting solution was washed with distilled water several times after dilution with dichloroethane. After that, the combined organic phase was dried with anhydrous $Na_2SO_4$ overnight. Then, the solvent was removed through rotary evaporation to

get a reddish-brown solid. (6.2 g, yield 75.7%). mp: 94 ºC. $^1$H-NMR (400 MHz, DMSO-$d_6$, ppm): $\delta = 8.07$ (t, 1 H), 7.48 (d, 1 H), 7.43 (dd, 1 H), 7.11–7.00 (m, 2 H), 4.40–4.30 (m, 2 H), 4.26–4.16 (m, 1 H), 3.92–3.87 (m, 1 H), 3.85 (d, 3 H), 3.44–3.34 (m, 2 H), 2.85 (ddd, 2 H), and 2.71 (ddd, 2 H). $^{13}$C-NMR (100 MHz, DMSO-$d_6$, ppm): $\delta = 158.85$, 149.15, 148.94, 143.53, 130.88, 129.46, 127.38, 118.71, 113.18, 109.38, 69.80, 55.61, 52.98, 49.64, 49.06, 45.03, and 43.79. FT-IR (KBr, cm$^{-1}$): 3400 (O–H, N–H), 1650 (C = O), and 905 (epoxy).

**Preparation of the cured epoxy resin.** GSPZ-EP cured with DDM as a curing agent was also studied in this work. The ratio of GSPZ-EP to DDM was fixed according to the stoichiometry (the molar ratio of the epoxy group to N–H was 1:1). GSPZ-EP was molten at 110 ºC, then the stoichiometric amount of DDM was rapidly added, and the mixture was vigorously stirred, until a homogeneous mixture was obtained. Afterward, it was degassed in a vacuum oven at 110 ºC to remove entrapped air. The gas-free mixture was then poured into a preheated stainless-steel mould, followed by curing at 80 ºC for 2 h, 150 ºC for 2 h, and 200 ºC for 2 h in a muffle furnace to get a completely cured resin, which was named as GSPZ-EP/DDM. As a comparison, DGEBA was cured with DDM in the same way to obtain a cured DGEBA/DDM resin.

**Characterization.** $^1$H-NMR, $^{13}$C-NMR, and HMBC spectroscopy were performed on the 400 MHz Bruker nuclear magnetic resonance spectrometer at room temperature using deuterated dimethylsulfoxide (DMSO-$d_6$) as solvent and tetramethylsilane as internal standard.

Fourier transform-infrared (FT-IR) spectra were recorded using a Thermo Nicolet Nexus 470 FT-IR spectrometer in the 400–4000 cm$^{-1}$ region.

Electrospray ionization mass spectroscopy (ESI-MS) was performed with an Ultimate XB-C18 column in liquid chromatography mode. Samples were analyzed by selected ion monitoring using ESI negative ion mode (CID = 50 V).

The rheological behavior of GSPZ-EP was investigated using a TA AR2000 instrument having a 25-mm-diameter parallel plate geometry with a 1000-μm gap. The temperature was swept from 25 to 275 ºC with a heating rate of 3 ºC·min$^{-1}$ under a frequency of 1 Hz and a stress of 10 Pa. The bulk viscosities of GSPZ-EP and DGEBA were measured using a Brookfield Viscometer (DV-II + Pro) at 110 ºC.

The curing behavior of the different epoxy resin systems was studied on a Mettler DSC1 differential scanning calorimeter under a nitrogen flow of 50 mL·min$^{-1}$. For the self-curing system, about 5 mg of GSPZ-EP was enclosed in an aluminum crucible. A heat scan ranging from 25 to 300 ºC was performed at heating rates of 5, 10, 15, and 20 ºC·min$^{-1}$. For the GSPZ-EP/DDM and DGEBA/DDM systems, the epoxy-curing mixtures were prepared as follows. GSPZ-EP was molten at 110 ºC, then the stoichiometric amount of DDM (epoxy group: N–H = 1:1 by mole) was quickly added, and vigorously stirred, until a homogeneous curing mixture was obtained. The curing mixture of DGEBA/DDM was prepared in the same way. About 5 mg of the as-prepared curing mixture was enclosed in an aluminum crucible and heated from 25 to 250 ºC. Four heating rates of 5, 10, 15, and 20 ºC·min$^{-1}$ were registered on each fresh sample.

TGA-DSC simultaneous thermal analysis was performed on a TA Q600 instrument at a heating rate of 10 ºC·min$^{-1}$ using about 5 mg of sample over a temperature range of 30–800 ºC in $N_2$ atmosphere.

Thermal gravimetric analysis (TGA) was performed on a Mettler TGA 1 instrument at a heating rate of 20 ºC·min$^{-1}$ over a temperature range of 30–800 ºC in $N_2$ and air.

Dynamic mechanical analysis (DMA) was carried out with a TA Q800 instrument at 1 Hz and a heating rate of 3 ºC·min$^{-1}$ from 25 to 230 ºC under air atmosphere. The rectangular specimens with dimensions of $35 \times 5.3 \times 3.0$ mm were fastened on a single cantilever clamp.

Tensile properties were tested according to GB/T 1040–2006 on a WSM-50KN electric universal testing machine (Intelligent Instrument Equipment, Changchun, China). Samples of a type 1BA geometry were tested with a cross-head speed of 5 mm·min$^{-1}$. The results were averaged from five measurements.

Microscale combustion calorimetry (MCC) was performed on FTT0001 microscale combustion calorimeter (UK), according to ASTM D7309–13. About 5 mg of cured epoxies were heated from 80 to 700 ºC, at a heating rate of 10 ºC·min$^{-1}$ with a nitrogen flow of 80 cm$^3$·min$^{-1}$. Then, the anaerobic and volatile products were mixed with a stream flowing at 20 cm$^3$·min$^{-1}$, and the stream included 80% nitrogen and 20% oxygen.

UL-94 vertical burning tests were carried out on a M607B-type instrument (Qingdao Shanfang Instrument Co., Ltd., Shandong Province, China) on the specimens of $125 \times 13 \times 3$ mm according to ASTM D3801–10.

## Data availability

The authors declare that the data in this study are available from the corresponding authors upon reasonable request.

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

## Acknowledgements

This work was supported by the National Nature Science Foundation of China (nos. 51873027, U1663226, and 51673033) and the Fundamental Research Funds for the Central Universities (DUT17LK39). We also thank Prof. Tao Tang in the Changchun Institute of Applied Chemistry, Chinese Academy of Sciences for assistance with the MCC test.

## Author contributions

Y.Q., Z.W. and X.J. proposed the project, designed and conducted the experiments, and wrote the paper. J.W., Y.K. and S.Z. proposed the project, H.P., N.L., and C.L. contributed reagents/materials/analysis tools.

## Additional information

**Competing interests:** The authors declare no competing interests.

