## [Peer Review File · Nature Communications]

Reviewers' comments:

Reviewer #1 (Remarks to the Author):

The manuscript (NCOMMS-18-28611) reported the four-step synthesis of epoxy resin monomer GSPZ-EP from biomass-derived guaiacol, succinic and epichlorohydrin (EP). After curing with DDM, the GSPZ-EP presented a higher T_g (by 140°C), storage modulus (by 70%) and Young's modulus (by 93%) in comparison to DEGBA. Additionally, the intrinsic flame retardancy of GSPZ-EP/DDM was obtained with a strong tendency of self-extinguishment in UL-94 vertical burning test. The authors performed a deep investigation and analysis of the curing kinetics for proposing the curing and self-curing mechanism. However, in the reviewer's opinion, even though the biomass-derived sources were employed, large amounts of organic solvents and complicated steps (four steps) were involved in the synthesis process. Compared with previous reports (ACS Sustainable Chemistry & Engineering, 2016, 4, 2869–2880; Journal of Materials Chemistry A, 2015, 3, 21907–21921, etc., which were already published), the enhancement of properties (T_g, mechanical property and flame retardancy) was not adequately outstanding. In particular, the organization of this manuscript shared a lot of similarities with the published. In the manuscript, the authors put emphasis on the improvement of flame retardancy of GSPZ-EP/DDM relative to DEGBA/DDM, nevertheless without a clear molecule-level interpretation of the underpinning carbonization mechanism, which was critical to the intrinsic flame retardancy.

The incorporation of higher aromatic structure gave rise to higher char yield (in N₂), better flame retardancy and higher modulus, while less segment mobility and lower elongation at break. Therefore, to some extent, it was difficult to conclude whether the mechanical property was excellent or not, particularly in consideration of the brittleness feature of epoxy resin matrix. It was unacceptable in the manuscript that the authors incorrectly evaluated the UL-94 rating as well. In the XPS analysis of the char morphology of GSPZ-EP/DDM and DEGBA/DDM, the higher content of N relative to C did not infer the better char morphology. The above assessment pushed the reviewer to reject the manuscript for publication in Nature Communications.

Reviewer #2 (Remarks to the Author):

The authors demonstrate a novel synthetic route to the fully biobased aromatic N-heterocyclic compounds, GSPZ, which they convert by traditional methods to a di-epoxy resin 'GSPZ-EP'. The authors mention its low viscosity, this should be emphasized as an important characteristic of this epoxy resin as it will allow for infusion into fiber matrices. The authors describe the excellent properties of cured resins with DDM relative to DGEBA-DDM. The demonstrated properties of this new biobased epoxy based system is very impressive. They also discuss rather convincingly that the synthesis of GSPZ-EP has excellent potential for future scale-up given the relative simplicity of individual reactions. The flame retardancy discussion is far too detailed as discussed below. What the authors failed to look at is the potential that GSPZ is an endocrine disruptor. This is critically important before moving forward with this molecule. Material characterizations are well done. In the opinion of this Reviewer, this paper would be better placed in Advanced Materials that has an impact factor of 21.

1) Self-curing behavior and proposed self-curing mechanism of GSPZ-EP., lines 134-136: As mentioned above, a tertiary amine represents a class of effective curing agents for epoxy resins, therefore, the curing reaction of GSPZ-EP may take place without the aid of extra curing agent. Reviewer comment: As written this is confusing. DDM is the curing agent and the extent of cure, crosslinking and molecular weights of products formed in the absence of DDM is not well characterized. In the presence of DDM, the tertiary amine is functioning as a curing agent catalyst.

2) Page 7, lines 185-187: This was probably due to the ring nitrogen attack on the epoxide to form

a zwitterion. Since this process was not a cross-linking reaction, the steric hindrance and the viscosity of the system would not be affected.

Reviewer comment: Assuming the ring-opening due to the ring nitrogen attack on the epoxide occurs as described in Figure 3e, the viscosity should increase as the molecular weight of the conjugate increases. You would essentially be forming dimeric species from the first reaction.

3)Page 7, lines 192-194: During the course of the reaction, the steric hindrance and viscosity increase as the network is formed requiring a higher activation energy as the reaction continues. This interpretation of the isoconversional analysis data further verifies that the mechanism reported in Figure 3(e) is reasonable.

Reviewer comment: The authors describe a network structure. Given this assertion, they should estimate the molecular weight between crosslinks of this system.

4)Reviewer comment: The results and discussion on char formation should be shortened to one or two paragraphs that summarize the essential learnings from the work carried out. The more detailed discussion could move to the supplementary information section. Alternatively, this could be the subject of a separate manuscript.

5)Reviewer comment: In numerous locations in the manuscript the authors say motivation when I believe they mean mobility.

6)Reviewer comment: Tensile strength is an abbreviated version for ultimate tensile strength. The authors should use the latter terminology.

7)Reviewer comment: Its unclear why the authors are citing crosslink density values from Ref 58 since the cited paper uses a very different epoxy monomer.

Reviewer #3 (Remarks to the Author):

The work deals work design and synthesis of a new pyridazine-based compound, namely, 6-(4-hydroxy-3-methoxyphenyl)pyridazin-3(2H)-one (GSPZ) by making use of guaicol and succinic anhydride as the starting materials. Notably, both the starting materials are derived from bio-mass which is an attractive feature of the work. GSPZ was converted into epoxy resin by reaction with bio-based epichlorohydrin. The curing behaviour of GSPZ-EP with petro-based diamine, namely, 4,4'-diaminodiphenyl methane was studied and thermal and mechanical properties of cured resins were evaluated. A comparison of the data was made with conventional petroleum-based bisphenol A epoxy resin.

The work has been planned well and executed meticulously. The results have been interpreted reasonably well in light of the previous studies. The manuscript is recommended for publication in Nature Communications

The following are the observations:

1. It would be good to compare bulk viscosity of GSPZ-EP with that of DGEBA
2. The rationale for a choice of 4,4'-diaminodiphenyl methane as the curing agent need to be explained.

Reviewer #1 (Remarks to the Author):

Comments:

The manuscript (NCOMMS-18-28611) reported the four-step synthesis of epoxy resin monomer GSPZ-EP from biomass-derived guaiacol, succinic and epichlorohydrin (EP). After curing with DDM, the GSPZ-EP presented a higher Tg (by 14°C), storage modulus (by 70%) and Young's modulus (by 93%) in comparison to DEGBA. Additionally, the intrinsic flame retardancy of GSPZ-EP/DDM was obtained with a strong tendency of self-extinguishment in UL-94 vertical burning test. The authors performed a deep investigation and analysis of the curing kinetics for proposing the curing and self-curing mechanism.

1. However, in the reviewer's opinion, even though the biomass-derived sources were employed, large amounts of organic solvents and complicated steps (four steps) were involved in the synthesis process.

Response: We thank the reviewer for this comment. Yes, organic solvents are still necessary for synthesizing bio-based compounds and polymers, as the solvents derived from biomass are particularly rare. Although total three steps were involved in the preparation of bio-based aromatic *N*-heterocycle (four steps for epoxy precursor), they were facile and straightforward, including Friedel-Crafts acylation, cyclization and dehydrogenation. Moreover, the products were easily isolated without utilizing column chromatography, implying potential for scale up or commercial applications. Up to now, synthesis of aromatic *N*-heterocyclic compounds from biomass-derived starting materials remains challenging and is still very limited, owing to the lack of efficient methods for biomass-derived precursors. For example, as mentioned in the main text, Bhusal *et al.* (*Green Chem.*, 2016, 18, 2453-2459) have synthesized a suit of diverse *N*-heterocycles from dimethyl itaconate and pyrrole, two compounds attainable from biomass (where the pyrrole was produced from furan and ammonia). And in this case, the reaction condition was relatively harsh and difficult to scale up.

2. Compared with previous reports (ACS Sustainable Chemistry & Engineering, 2016, 4, 2869–2880; Journal of Materials Chemistry A, 2015, 3, 21907-21921, etc., which were already published), the enhancement of properties (Tg, mechanical property and flame retardancy) was not adequately outstanding. In particular, the organization of this manuscript shared a lot of similarities with the published.

Response: Many thanks for the reviewer. The two previous reports (ACS Sustainable Chemistry & Engineering, 2016, 4, 2869–2880; Journal of Materials Chemistry A, 2015, 3, 21907-21921) mentioned by reviewer actually have been recited in the main text as Ref 51 and 55, respectively. We really appreciated the authors of these references, we have learned a lot from their work, and also hope to make progress on the basis of their work. As shown in Figure R1(b, c), both of these two references reported the high performance bio-based epoxy resins based on eugenol, the other sources were cyanuric chloride (Ref 51) and terephthaloyl (Ref 55), respectively, which both were derived from petroleum (marked with red color in Figure R1). Therefore, the carbon sources of the as-prepared

epoxy resin in Ref 51 and 55 were partially bio-based, whereas the GSPZ-EP in our work was fully derived from bio-based carbon. Moreover, the curing agent employed in Ref 51 and 55 both was 3,3'-diamino diphenyl sulfone (33DDS), and its melting point (170-173 °C) was much higher than that of 4,4'-Diaminodiphenyl methane (DDM, 90 °C) employed in this work. Usually, under the identical curing conditions, the epoxy resin cured with 33DDS can get higher thermal and mechanical properties than that with DDM. As summarized in Table R1, the properties of GSPZ-EP/DDM were competitive over those reported in Ref 51 and 55. To our best knowledge, the values of T_g and char yield at 800 °C in our work were in the top rank of the fully bio-based epoxy resin. And about flame retardancy, as mentioned in Ref 55, after ignition in 10 s, TPEU-EP/33DDS extinguished at about 24 s, and they couldn't ignite that sample again by using the burner for 10 s. For GSPE-EP/DDM in this work, the sample extinguished at about 5.3 s and 7 s, respectively, after first and second application of the flame, which illustrated that the sample had good intrinsic flame retardancy. Again, it should be pointed out that GSPZ is the first aromatic *N*-heterocyclic monomer as far as we know, where all the carbon sources were derived from biomass. And we believe that this facile and straightforward strategy for synthesizing bio-based aromatic *N*-heterocycle can supply some valuable information for the audiences of Nature Communications.

Figure R1 Molecular structures (the petroleum sources were marked with red color) comparison of epoxy resins and curing agents.

Table R1 Properties comparison of bio-based epoxy resins.

Epoxy resin	T_g^a (°C)	Char yield at 800 °C (%)	E' at 30 °C (MPa)	Tensile properties		Ref
				Strength (MPa)	Modulus (GPa)	
GSPZ-EP/DDM	187	40.4	3327	63.5	2798	This work
TEU-EP/33DDS	207	25.1	~3750	NA	NA	51
TPEU-EP/33DDS	168.4	31.7	3470	NA	NA	55

^a T_g : The values were obtained from the peak value temperature of $\tan\delta$ against temperature curves from DMA tests.

51. Wan, J. *et al.* Ultrastiff biobased epoxy resin with high T_g and low permittivity: From synthesis to properties. *ACS Sustain. Chem. Eng.* **4**, 2869-2880 (2016).

55. Wan, J. T. *et al.* A novel biobased epoxy resin with high mechanical stiffness and low flammability: synthesis, characterization and properties. *J. Mater. Chem. A.* **3**, 21907-21921 (2015).

3. In the manuscript, the authors put emphasis on the improvement of flame retardancy of GSPZ-EP/DDM relative to DEGBA/DDM, nevertheless without a clear molecule-level interpretation of the underpinning carbonization mechanism, which was critical to the intrinsic flame retardancy.

Response: We thank the reviewer for this comment. We would like to make response in conjunction with *Question 6* (Reviewer #1) and *Question 4* (Reviewer #2). In the Section of Flame retardancy and mechanism of cured epoxy resins (main text), we have conducted several measurements to explain the mechanism of flame retardancy and carbonization, including the SEM and XPS analyses of the residual chars, and 3D TG-IR spectra of the thermal degradation component in gas phase for the as-prepared epoxy resins. And in the revision, we also added some discussion based on the XPS data, to show that more chars with steady cross-linked structures composed with C=C and C=N were formed by aromatization at high temperature in GSPZ-EP/DDM system. Moreover, the tertiary amine in GSPZ-EP could not only promote charring but also participate in charring to become part of the char residues (which can also result in higher content of N relative to C), and this can lead to the formation of the dense and stronger char layer structure after combustion, showing better flame retardancy. We hope the additional information can help to illustrate the carbonization mechanism more clearly, as well as answer the *Question 6* –“In the XPS analysis of the char morphology of GSPZ-EP/DDM and DEGBA/DDM, the higher content of N relative to C did not infer the better char morphology.”

However, as suggested by Reviewer #2 in *Question 4* ---“The results and discussion on char formation should be shortened to one or two paragraphs that summarize the essential learnings from the work carried out. The more detailed discussion could move to the supplementary information section. Alternatively, this could be the subject of a separate manuscript.”, we have refined the relevant content. the SEM images of the residual chars (Figure 6) and the 3D TG-IR spectra of the degradation component in gas phase (Figure 7), as well as the corresponding detailed discussion (including XPS analysis) has been moved to the supplementary information section. For more detailed information, please see the Page 11 in the main text, as well as Page 13-17 in Supplementary Information section. Because of the large changes and more content, they are no longer listed here. We apologized for the inconvenience.

According to the comment, the discussion on XPS data has been modified as follows (transferred from main text to Supplementary Information):

Supplementary Information section Page 16: *The chemical composition and states of the char residues after UL-94 test were further analyzed by XPS. The details are shown in Fig S13. The C1s spectrum of charred residues collected from GSPZ-EP/DDM can be*

split into more peaks than DGEBA/DDM. The peaks at 284.1 eV is ascribed to C-C and C-H in aliphatic and aromatic groups, the peak at 285.4 eV can be assigned to C-N incyclized compounds^{S4}, and the peak at 291 eV is attributed to the shake up satellite of π - π^* transition^{S5}. All the above three peaks can be found in both GSPZ-EP/DDM and DGEBA/DDM spectra. However, the peak centered at 287.5 eV attributed to C=N and C=C^{S6} groups can only be observed in the C1s spectrum of GSPZ-EP/DDM, which indicated more chars with steady cross-linked structures composed with C=C and C=N were further formed by aromatization at high temperature in GSPZ-EP/DDM system. Generally, aromatic structure formed during the degradation process playing an important role on self-extinguish and flame retardancy during combustion^{S7}. The N1s spectrum of DGEBA/DDM shown in Fig.S13(e) can be split into only one peak centered at 399.3eV, which is attributed to pyrrole nitrogen. While for GSPZ-EP/DDM, besides the peaks attributed to pyrrole nitrogen, it can also be split into another peak centered at 398.9 eV attributed to pyridine nitrogen^{S8}, which is more thermal stable and benefit to form more stable carbon layer^{S9-11}. These observations are perhaps due to the tertiary amine in the pyridazine structure of GSPZ-EP. The tertiary amine could not promote charring but also participate in charring to become part of the char residues. What' more, it could also make the char layer more stable, because more stable char structure such as C=C, C=N and pyridine nitrogen structure exist in the char residues of GSPZ-EP/DDM system. The same conclusion could also be drawn from Table S1: after combustion there is more nitrogen present in the charred residues of GSPZ-EP/DDM (1.3 times higher than that of DGEBA/DDM). These results also indicate nitrogen in GSPZ-EP could participate in charring, leaving more nitrogen in the residual char. All these could benefit to promote both the quality and quantity of the residue char, leading to the formation of the dense and stronger char layer structure after combustion.

4. The incorporation of higher aromatic structure gave rise to higher char yield (in N₂), better flame retardancy and higher modulus, while less segment mobility and lower elongation at break. Therefore, to some extent, it was difficult to conclude whether the mechanical property was excellent or not, particularly in consideration of the brittleness feature of epoxy resin matrix.

Response: Many thanks for the useful suggestions from our reviewer. Considering the lower elongation at break in GSPZ-EP/DDM compared to DEGBA/DDM (Figure 6b), it is not easy to say the mechanical property was excellent for GSPZ-EP/DDM. According to the comment, we deleted the sentence 'To summarise, GSPZ-EP/DDM shows excellent mechanical properties.' in the main text.

Page 13 Line 2: However, the *ultimate* tensile strength of GSPZ-EP/DDM is still very high.

5. It was unacceptable in the manuscript that the authors incorrectly evaluated the UL-94 rating as well.

Response: Many thanks for the reviewer, and sorry for the incorrect evaluation of the UL-94 rating. Actually, we conducted UL-94 vertical burning test according to ASTM D3801-10, and the criteria conditions were list in Table R2. The results of our sample can

meet all the requirements of V0 rating, except the total afterflame time for any condition set (t_1 plus t_2 for the five specimens) (Second line in Table R2). For this item, the value in our case was less 70 s, which was little higher than the criterion of V0 (≤ 50 s) but much lower that of V1 (≤ 250 s). Therefore, we would like to carefully say that the cured GSPZ-EP/DDM showed good intrinsic flame retardancy properties and was very close to the V0 rating of the UL-94 test. And the description and discussion in the corresponding main text have been revised as follows:

Abstract Line 12: *Moreover, the cured GSPZ-EP/DDM shows **good** intrinsic flame retardancy properties and **is very close to the V-0 rating of the UL-94 test.***

Page 10 Line 6: *Following a second ignition, after 7 s, it still could self-extinguish without any melt-dripping. **These results indicated that the intrinsic flame retardancy property of cured GSPZ-EP/DDM was good and very close to the highest UL-94 rating of V0.***

Page 13 last Line: *Furthermore, it **was very close to the highest UL-94 rating of V0**, while DGEBA/DDM **passed** no rating.*

Table R2 Materials classification^a

Criteria Conditions	V-0	V-1	V-2	Our Work
Afterflame time for each individual specimen, t_1 or t_2	≤ 10 s	≤ 30 s	≤ 30 s	<10 s
Total afterflame time for any condition set (t_1 plus t_2 for the five specimens)	≤ 50 s	≤ 250 s	≤ 250 s	<70 s
Afterflame plus afterglow time for each individual specimen after the second flame application ($t_1 + t_2$)	≤ 30 s	≤ 60 s	≤ 60 s	<30 s
Afterflame or afterglow of any specimen up to the holding clamp	No	No	No	No
Cotton indicator ignited by flaming particles or drops	No	No	Yes	No

^a If only one specimen from a set of five specimens does not comply with the requirements, another set of five specimens shall be tested. In the case of the total number of seconds of flaming, an additional set of five specimens shall be tested if the totals are in the range from 51 to 55 s for V-0 and from 251 to 255 s for V-1 and V-2. All specimens from this second set shall comply with the appropriate requirements in order for the material in that thickness to be classified V-0, V-1, or V-2.

6. In the XPS analysis of the char morphology of GSPZ-EP/DDM and DEGBA/DDM, the higher content of N relative to C did not infer the better char morphology.

Response: Many thanks for the comment. As mentioned in the response section to *Question 3* (Reviewer #1), some additional information has been added in the discussion of XPS data, thus we would like to carefully say that the higher content of N relative to C can infer the better char morphology. Because the tertiary amine in the pyridazine of GSPZ-EP could not only promote charring but also participate in charring to become part of the char residues. Moreover, it could also make the char layer more stable, as more

sable char structure such as C=C, C=N and pyridine nitrogen structure exist in the char residues of GSPZ-EP/DDM system. The changes of the content on XPS discussion can be seen in the response section to *Question 3* (Reviewer #1), as well as Page 11 in the main text and Page 16 in Supplementary Information section, we do not repeat that here.

Reviewer #2 (Remarks to the Author):

The authors demonstrate a novel synthetic route to the fully biobased aromatic N-heterocyclic compounds, GSPZ, which they convert by traditional methods to a di-epoxy resin 'GSPZ-EP'. The authors mention its low viscosity, this should be emphasized as an important characteristic of this epoxy resin as it will allow for infusion into fiber matrices. The authors describe the excellent properties of cured resins with DDM relative to DGEBA-DDM. The demonstrated properties of this new biobased epoxy based system is very impressive. They also discuss rather convincingly that the synthesis of GSPS-EP has excellent potential for future scale-up given the relative simplicity of individual reactions. The flame retardancy discussion is far too detailed as discussed below.

Response: We really appreciate the reviewer for the positive comments for this manuscript.

What the authors failed to look at is the potential that GSPZ is an endocrine disruptor. This is critically important before moving forward with this molecule. Material characterizations are well done. In the opinion of this Reviewer, this paper would be better placed in *Advanced Materials* that has an impact factor of 21.

Response: We thank the reviewer for the overall positive evaluation of our work. Yes, usually, the bisphenol analogs are the potential endocrine disruptor. Recently, some reports (*Green Chem.*, **18**, 4961-4973 (2016); *Int. J. Environ. Res. Public Health*, **13**, 705-720 (2016); *Endocrinology*, **147**, 4132-4150 (2006).) have illustrated that both the sites *ortho* to the phenolic hydroxyl substituted with methoxy moieties and the addition of methoxy groups around the aromatic ring can significantly decrease binding affinity to endocrine compound. And for GSPZ, the *ortho* position to the phenolic hydroxyl was substituted with methoxy group, based on the previous studies, we would like to carefully say that GSPZ maybe is not a serious endocrine disruptor.

1. Self-curing behavior and proposed self-curing mechanism of GSPZ-EP., lines 134-136: As mentioned above, a tertiary amine represents a class of effective curing agents for epoxy resins, therefore, the curing reaction of GSPZ-EP may take place without the aid of extra curing agent.

Reviewer comment: As written this is confusing. DDM is the curing agent and the extent of cure, crosslinking and molecular weights of products formed in the absence of DDM is not well characterized. In the presence of DDM, the tertiary amine if functioning as a curing agent catalyst.

Response: Sorry for the confusing writing. Actually, as reported by previous reports (main text Ref 33-35), tertiary amine can act as *catalytic* curing agent which can lead to epoxy homopolymerization. In our case, we have tested the thermal stabilities of the self-cured

GSPZ-EP without extra curing agent (curing process: 80°C-2h+150°C-2h+200°C-2h, which was identical to that in GSPZ/DDM system). As shown in Figure R2 and Table R3, even the TGA data of the self-cured GSPZ-EP without DDM were similar to those of GSPZ-EP/DDM system (Table 1 in main text), the T_g of the former (120°C via DSC) was much lower than that of the latter (187°C via DMA). Therefore, we just focused on the curing procedure and properties of GSPZ-EP/DDM system, and that also was the main reason why we did not supply the information about the extent of cure, crosslinking and molecular weights of products formed in the absence of DDM as the reviewer asked.

Yes, the tertiary amine also can act as a curing agent catalyst even in the presence of DDM. The GSPZ-EP/DDM system presented lower activation energy (48.0 kJ/mol) than DGEBA/DDM (53.6 kJ/mol), which may be due to the presence of the tertiary amine group. This also has been confirmed by previous report. For example, as shown in Figure R3, Wan *et al.* have found that compared to PDA, TAPA can lead to the lower effective activation energy due to its catalytic tertiary amine groups for curing DGEBA (Chem. Eng. J., 2012, 188:160-172)).

According to the comments, we revised the manuscript as follows:

Page 5 Line 3: *These are an effective class of catalytic curing agent widely used to initiate polymerization of various epoxy resins*³³⁻³⁵.

Page 5 Line 12: *As mentioned above, a tertiary amine represents a class of effective catalytic curing agents for epoxy resins, therefore, we employed TGA-DSC simultaneous thermal analysis to track the reaction process without curing agent (Figure 3(a)).*

Figure R2 Thermal stabilities of self-curing GSPZ-EP without extra curing agent. (a) TGA curve in N₂; (b) TGA curve in air; (c) DSC curve.

Table R3 Key parameters of thermal stabilities of self-curing GSPZ-EP

Sample	T _{d5%} (°C)		C _{yd700} (%)		T _{max} (°C)		R _{max} (%/min)	
	N ₂	Air	N ₂	Air	N ₂	Air	N ₂	Air
GSPZ-EP	323	316	45.8	0.6	379	389、605	0.1	0.07, 0.11

Figure R3 Molecular structure of curing agent (Chem. Eng. J., 2012, 188:160-172).

2. Page 7, lines 185-187: This was probably due to the ring nitrogen attack on the epoxide to form a zwitterion. Since this process was not a cross-linking reaction, the steric hindrance and the viscosity of the system would not be affected.

Reviewer comment: Assuming the ring-opening due to the ring nitrogen attack on the epoxide occurs as described in Figure 3e, the viscosity should increase as the molecular weight of the conjugate increases. You would essentially be forming dimeric species from the first reaction.

Response: Thanks for the comment from reviewer. According to the curing mechanism of epoxy-tertiary amine systems reported by previous references (main text Ref. 41-42), dimeric species should be formed in the first reaction. And we also tried the corresponding experiment to confirm this in our system. For details, the GSPZ-EP was heated to 200 °C (the exothermic peak temperature of the first reaction in Figure 3(a)) on DSC at a heating rate of 20 °C/min, after the sample was cooled down to room temperature, it can be completely dissolved in dimethyl sulfoxide (DMSO), then the molecular weight of the resulting product was analyzed via MALDI-TOF MS. As shown in Figure R4, after heated GSPZ-EP to 200 °C quickly without heat preservation, the main product were dimeric species. As comments from reviewer, the viscosity should increase as the molecular weight of the conjugate increase. However, as we know, the higher temperature will result in lower viscosity of the epoxy precursor. We assumed that the effects of temperature and molecular weight on the viscosity of the system offset each other, therefore, as shown the rheological measurement in Figure 3(b) (main text), the viscosity of GSPZ-EP remained approximately constant before the cross-linking reaction occurred. Anyhow, we have made modification in the main text as follows:

Page 7 Line 22: *For the first exothermic peak, the activation energy remained constant for the entire range of conversion, similar results also were observed by Leena et al.³⁸.*

Figure R4 MALDI-TOF MS spectrum of the product formed by heating GSPZ-EP to 200 °C on DSC at a heating rate of 20 °C/min.

3. Page 7, lines 192-194: During the course of the reaction, the steric hindrance and viscosity increase as the network is formed requiring a higher activation energy as the reaction continues. This interpretation of the isoconversional analysis data further verifies that the mechanism reported in Figure 3(e) is reasonable.

Reviewer comment: The authors describe a network structure. Given this assertion, they should estimate the molecular weight between crosslinks of this system.

Response: We really appreciate the reviewer's suggestion. The molecular weight between crosslinks can be calculated based on the crosslinking density, and the detailed calculation method of the crosslinking density can be found in the *Response to Question 7*. Unfortunately, we failed to get the non-defect sample for DMA measurement to get the storage modulus E' and T_g for calculating crosslinking density. Therefore, we can not supply the information about the molecular weight between crosslinks herein. However, as mentioned in the *Response to Question 1*, the T_g of self-cured GSPZ-EP without DDM (120°C via DSC) was much lower than that of GSPZ-EP/DDM (187°C via DMA) under the identical thermal conditions, then we turned our attention to the latter system, and the structure and properties of self-cured GSPZ-EP were not studied very well.

4. Reviewer comment: The results and discussion on char formation should be shortened to one or two paragraphs that summarize the essential learnings from the work carried out. The more detailed discussion could move to the supplementary information section. Alternatively, this could be the subject of a separate manuscript.

Response: We really appreciate the reviewer's suggestion. We have refined the relevant content, for example, the SEM images of the residual chars (Figure 6) and the 3D TG-IR spectra of the degradation component in gas phase (Figure 7), as well as the corresponding detailed discussion (including XPS analysis) has been moved to the supplementary information section. For more detailed information, please see the Page 11-13 in the main text, as well as Page 13-17 in Supplementary Information section. Because of the large changes and more content, they are no longer listed here. We apologized for the inconvenience.

5. Reviewer comment: In numerous locations in the manuscript the authors say motivation when I believe they mean mobility.

Response: Thank you very much for your careful review, and sorry for our typo. We have checked the manuscript carefully, and replaced 'motivation' by 'mobility' in the text as follows:

Page 12 line 18: and restricts the molecular chain *mobility*, which is in favor of improving the T_g of the system.

Page 12 line 30:at the same time restricting the *mobility* of molecular chain,....

6. Reviewer comment: Tensile strength is an abbreviated version for ultimate tensile strength. The authors should use the latter terminology.

Response: Thank you very much for your kind suggestion. According to the comment, we have replaced the word 'tensile strength' by 'ultimate tensile strength' in the text.

Page 13 line 1: it doesn't show obviously improvement in *ultimate* tensile strength.

Page 13 line 2: However, the *ultimate* tensile strength of GSPZ-EP/DDM is still very high.

Page 16 Table 2:

Table 2 Comparison of properties from DMA and tensile testing for GSPZ-EP/DDM and DGEBA/DDM

Sample	E' (30 °C) ^a (MPa)	T _g ^b (°C)	V _r ^c (mol/m ³)	Young's modulus (MPa)	Ultimate tensile strength (MPa)	Elongation at break (%)
GSPZ-EP/DDM	3327±38	187	1200±68	2798±120	63.5±5	2.8±0.5
DGEBA/DDM	1952±20	173	2432±46	1452±108	61.7±3	6.3±0.3

^a E'(30 °C): Storage modulus at 30 °C measured by DMA.

^b T_g: Glass transition temperature defined by the peak value temperature of tanδ against temperature curve.

^c V_r: Cross-linking density.

7. Reviewer comment: Its unclear why the authors are citing crosslink density values from Ref 58 since the cited paper uses a very different epoxy monomer.

Response: We thank the reviewer for this comment. In this work, we calculated the cross-linking density according to the popular rubber elasticity theory (eqn 1) as mentioned in Ref 55 (It is Ref 58 in original manuscript), although the molecular structure of GSPZ-EP was different with that in Ref 55. S. Mitra *et al.* (*Prog. Org. Coat.*, **32**, 235-243 (2014)) have mentioned that the elastic modulus is independent of the chemical structure of the network, thus there have been numerous studies reported on application of the above theory for the rubbery region of crosslinked networks such as epoxy and polyester system. Therefore, the eqn 1 also was employed directly to calculate the cross-linking density of our system.

$$E_r = 3RT_r V_r \quad (\text{eqn. 1})$$

Where, R is the universal gas constant, T_r is the temperature which is 30 °C above the glass transition temperature, E_r is the storage modulus obtained from DMA in the rubbery plateau region at (T_g+30°C) and V_r is the cross-linking density.

Reviewer #3 (Remarks to the Author):

The work deals work design and synthesis of a new pyridazine-based compound, namely, 6-(4-hydroxy-3-methoxyphenyl)pyridazin-3(2H)-one (GSPZ) by making use of guaiacol and succinic anhydride as the starting materials. Notably, both the starting materials are derived from bio-mass which is an attractive feature of the work. GSPZ was converted into epoxy resin by reaction with bio-based epichlorohydrin. The curing behaviour of GSPZ-EP with petro-based diamine, namely, 4,4'-diaminodiphenyl

methane was studied and thermal and mechanical properties of cured resins were evaluated. A comparison of the data was made with conventional petroleum-based bisphenol A epoxy resin.

The work has been planned well and executed meticulously. The results have been interpreted reasonably well in light of the previous studies. The manuscript is recommended for publication in Nature Communications.

Response: We thank the reviewer for his/her positive evaluation of the manuscript.

1. It would be good to compare bulk viscosity of GSPZ-EP with that of DGEBA.

Response: Thanks for the useful suggestion from our reviewer. Following the comment, we have added the bulk viscosity measurement in the Characterization Part, as well as the corresponding data in the main text.

Page 5 Line 9:curing reactivity than DGEBA/DDM under the same conditions. *Viscosity is an important parameter for epoxy resin, upon heating to 110 °C, the bulk viscosity for GSPZ-EP was 203 mPa·s, whereas it was 41 mPa·s in the case of DGEBA.*

Page 16 last second line :frequency of 1 Hz and a stress of 10 Pa. *The bulk viscosities of GSPZ-EP and DGEBA were measured using a Brookfield Viscometer (DV-□+Pro) at 110 °C.*

2. The rationale for a choice of 4,4'-diaminodiphenyl methane as the curing agent need to be explained.

Response: Thank you for your kind comment. 4,4'-Diaminodiphenyl methane is chosen as curing agent not only because of its high curing activity, but also because its melting point (90 °C) is similar to GSPZ-EP (92 °C), which is more conducive to the operation of curing reaction. Based on this, we added information in the corresponding text as follows:

Page 4 Line 11: *The non-isothermal curing behaviors of the bio- and petroleum-based epoxy resin systems were shown in Figure 2(a) with DDM as curing agent, Because its melting point (90 °C) is similar to GSPZ-EP (92 °C), which is more conducive to the operation of curing reaction.*

Reviewers' comments:

Reviewer #1 (Remarks to the Author):

The authors did some revisions according to the reviewers' comments. Some discussions became more clear which is good. As reviewer, I am sure this study has some values to the readers. However, as compared to the state of the art, it is hard to claim this study achieved a significant improvement. For example, in the revised version the study on the chabonization mechanism is still far way to a molecule-level interpretation. Furthermore, the author seems not very familiar with fire retardancy of materials and UL94 tests, because the V-0 rating is not the "highest rating" at all. It is somewhat irritating. The interpretation on fire and carbonization behaviours is strongly doubted.

Reviewer #2 (Remarks to the Author):

1) In Revision vs1, Reviewer 2 asked the following question and received an informative response. The contents of the response and any other relevant information should be included in the Introduction to justify the design, synthesis and studies of the synthesized epoxy resin.

What the authors failed to look at is the potential that GSPZ is an endocrine disruptor. This is critically important before moving forward with this molecule.

Response: We thank the reviewer for the overall positive evaluation of our work. Yes, usually, the bisphenol analogs are the potential endocrine disruptor. Recently, some reports (Green Chem., 18, 4961-4973 (2016); Int. J. Environ. Res. Public Health, 13, 705-720 (2016); Endocrinology, 147, 4132-4150 (2006).) have illustrated that both the sites ortho to the phenolic hydroxyl substituted with methoxy moieties and the addition of methoxy groups around the aromatic ring can significantly decrease binding affinity to endocrine compound. And for GSPZ, the ortho position to the phenolic hydroxyl was substituted with methoxy group, based on the previous studies, we would like to carefully say that GSPZ maybe is not a serious endocrine disruptor

Reviewer #2 made comments to the editor saying that a discussion about the endocrine disruptor possibility must be added.

Reviewer #1 (Remarks to the Author):

Comments:

The authors did some revisions according to the reviewers' comments. Some discussions became more clear which is good. As reviewer, I am sure this study has some values to the readers. However, as compared to the state of the art, it is hard to claim this study achieved a significant improvement. For example, in the revised version the study on the carbonization mechanism is still far way to a molecule-level interpretation.

Response: We really appreciate the reviewer for the comments on this manuscript. In the revised version, the relevant content about carbonization mechanism was moved to Supplementary Information according to the comment from previous Reviewer #2 in Question 4--"The results and discussion on char formation should be shortened to one or two paragraphs that summarize the essential learnings from the work carried out. The more detailed discussion could move to the supplementary information section. Alternatively, this could be the subject of a separate manuscript." Yes, maybe we need to do more to explain the carbonization at the molecular level. However, in this study, we would like to focus on the work to supply both a facile synthetic toolbox for sustainable production of aromatic *N*-heterocycles and a method to improve the competitiveness of bio-based epoxy resin relative to their petroleum-based counterpart. Therefore, we would like to remove all of the detailed discussion on carbonization mechanism from this manuscript, and consider this subject as a separate manuscript with our further work. The changes in main text and supplementary information have been made as follows:

Main text Page 10 Line 13: *These combined results indicate GSPZ-EP/DDM shows far superior flame retardancy to DGEBA/DDM, which may be attributed to the high charring ability aroused by the full aromatic structure with tertiary amine in former's network^{56,57}, and the proposed mechanism will be illustrated in our further work.*

Main text Page 10 Figure 5: *Figure 5(b) was deleted.*

Main text Page 11: *Both the first and second paragraphs were deleted.*

Main text Page 13 last sentence: ~~*SEM and XPS analysis indicated that post-combustion samples of GSPZ-EP/DDM could form intumescent strong charred layer structures, producing a flame retardancy mechanism. Meanwhile, the reduction in flammable gases released during the thermal degradation of GSPZ-EP/DDM, relative to the other polymer, could further enhance its flame retardancy.*~~

Main text Page 18: *The detailed information about the measurement methods for SEM, XPS and TG-IR was deleted.*

Supplementary Information: *Fig.S11, Fig.S12, Fig.S13, Table S1 and relevant references were deleted.*

Furthermore, the author seems not very familiar with fire retardancy of materials and UL94 test, because the V-0 rating is not the "highest rating" at all. It is

somewhat irritating.

Response: Thank you very much for the comment, and sorry for the confusion. Actually, what we want to say here is that V0 rating is the highest level in the UL-94 test method, not the highest level in all methods. For example, the description of fire retardancy rating in the main text is as follows: "These results indicated that the intrinsic flame retardancy property of cured GSPZ-EP/DDM was good and very close to the highest UL-94 rating of V0." Anyhow, we would like to delete the word "highest" in the text, and changes in main text have been made as follows:

Page 10 Line 7: *These results indicated that the intrinsic flame retardancy property of cured GSPZ-EP/DDM was good and very close to the ~~highest~~ UL-94 rating of V0.*

Page 13 last third Line: *Furthermore, it was very close to the ~~highest~~ UL-94 rating of V0, while DGEBA/DDM passed no rating.*

The interpretation on fire and carbonization behaviors is strongly doubted.

Response: We thank the reviewer for this comment. As mentioned in the above Response part, both the interpretation on carbonization behaviors and mechanism would be removed from this manuscript, and this subject will be considered as a separate manuscript with our further work.

Reviewer #2 (Remarks to the Author):

In Revision vs1, Reviewer 2 asked the following question and received an informative response. The contents of the response and any other relevant information should be included in the Introduction to justify the design, synthesis and studies of the synthesized epoxy resin.

What the authors failed to look at is the potential that GSPZ is an endocrine disruptor. This is critically important before moving forward with this molecule.

Response: We thank the reviewer for the overall positive evaluation of our work. Yes, usually, the bisphenol analogs are the potential endocrine disruptor. Recently, some reports (Green Chem., 18, 4961-4973 (2016); Int. J. Environ. Res. Public Health, 13, 705-720 (2016); Endocrinology, 147, 4132-4150 (2006).) have illustrated that both the sites ortho to the phenolic hydroxyl substituted with methoxy moieties and the addition of methoxy groups around the aromatic ring can significantly decrease binding affinity to endocrine compound. And for GSPZ, the ortho position to the phenolic hydroxyl was substituted with methoxy group, based on the previous studies, we would like to carefully say that GSPZ maybe is not a serious endocrine disruptor.

Reviewer #2 made comments to the editor saying that a discussion about the endocrine disruptor possibility must be added.

Response: Many thanks for the useful suggestions from our reviewer. According to the comments, we have added the relevant information about the endocrine disruptor possibility in the Introduction as follows:

Page 3 Line 18:introduction of the *pyridazine-based structure*. *Moreover, the*

methoxy moiety on the ortho site to phenolic hydroxyl in the as-prepared aromatic N-heterocycle can significantly decrease the binding affinity to endocrine compounds, whereas bisphenol A is considered as an endocrine disruptor because it can bind to a variety of receptors in biological systems³⁰⁻³³. This work affords both a facile synthetic toolbox.....

REVIEWERS' COMMENTS:

Reviewer #2 (Remarks to the Author):

The reviewers have made the needed changes to the manuscript. It is now suitable for publication

Reviewers' comments:

Reviewer #2 (Remarks to the Authors):

The reviewers have made the needed changes to the manuscript. It is now suitable for publication.

Response: Thank you very much for your positive evaluation, and it is great news for us.